# Can bone scintigraphy reflect the progression of osteoarthritis after unicompartmental knee arthroplasty?

**Sumin Lim**[1], **Tae Hun Kim**[1], **Do Young Park**[1,2], **Jong Min Lee**[1], **Jun Young Chung**[1]*

**1** Department of Orthopedic Surgery, Ajou University School of Medicine, Suwon, Korea, **2** Cell Therapy Center, Ajou University Medical Center, Suwon, Korea

* junyoungchung@gmail.com

## Abstract

### Background

Bone scintigraphy (BS) has been reported to be a useful predictor of osteoarthritis (OA) progression in primary knee OA. However, no previous studies have explored the relationship between BS and OA progression in the retained compartments after unicompartmental knee arthroplasty (UKA). Thus, we evaluated whether OA progresses to other compartments in patients who undergo UKA and if increased uptake on BS is associated with OA progression in other compartments after UKA.

### Methods

A total of 41 patients with knee BS at least five years after UKA were included. Radiographic OA progression in other compartments was assessed by grading and comparing OA severity in each patient using the Kellgren–Lawrence grading system (K-L grade) and Osteoarthritis Research Society International (OARSI) atlas score. After UKA, the correlation between BS uptake and radiographic OA progression was analyzed in each retained compartment. A correlation analysis was also performed to evaluate the association between BS uptake and OA progression grades.

### Results

A significant progression of OA was observed in both contralateral tibiofemoral and patellofemoral compartments after UKA at 98.5 ± 26.0 months of follow-up (all p<0.001). No correlation was found between BS uptake and radiographic OA progression nor between BS uptake and radiographic OA progression grade in the contralateral and patellofemoral compartments.

### Conclusions

Following UKA, OA progresses in the retained contralateral tibiofemoral and patellofemoral compartments over a minimum five-year follow-up period. Thus, BS is ineffective in assessing the progression of OA in these compartments.

**Data Availability Statement:** All relevant data are within the paper and its Supporting Information files.

**Funding:** The authors received no specific funding for this work.

**Competing interests:** The authors have declared that no competing interests exist.

## Introduction

Unicompartmental knee arthroplasty (UKA) is a widely accepted surgical treatment option for isolated medial or lateral compartment osteoarthritis (OA) of the knee. The use of UKA continues to increase, and long-term survival rates have been reported to be highly favorable, with 10-year survival rates of approximately 95% and 20-year survival rates of >90% [1–4]. However, there is a concern about the progression of OA after UKA as it preserves two additional compartments (the contralateral tibiofemoral and patellofemoral) after surgery, unlike total knee arthroplasty. Thus, evaluating OA progression in other compartments during the postoperative follow-up period is crucial. The primary causes of UKA failure are aseptic loosening and OA progression [5–9]. The imaging modality options for detecting such complications are narrow as lateral radiographs may show obscuration caused by the implant. Furthermore, magnetic resonance imaging (MRI) and computed tomography arthrography are limited by prosthetic interference [10–12].

Bone scintigraphy (BS) reflects alterations in the metabolic activity of the bone and has a unique capability of simultaneously demonstrating metabolically active joints throughout the body, and not just localized joint disease [13, 14]. The progression of OA is associated with high bone turnover, which in turn increases the potential for $^{99m}$Technetium-labelled methylene diphosphonate ($^{99m}$Tc-MDP) binding. Previously, BS was proven to be a predictor of knee OA progression [15–17]. However, the use of BS in patients who have undergone arthroplasty has usually been limited to assessing painful knee arthroplasty due to aseptic loosening or infection [18–21]. Since the knee joint after UKA contains two un-operated compartments, increased uptake on BS confined to un-operated compartments may not be explained solely by such painful complications.

Furthermore, an increased uptake on BS may reflect degenerative changes in the knee [16, 22, 23]; thus, it seems plausible to interpret the uptake in other compartments as a predictor of OA progression after UKA. To the best of our knowledge, no previous study has assessed the relationship between BS and OA progression in other compartments after UKA. Thus, we aimed to evaluate whether OA progresses in other compartments in patients undergoing UKA and whether increased uptake on BS is associated with OA progression in other compartments after UKA. We hypothesized that OA would progress with time and BS would be correlated with the radiographic progression of OA in other compartments after UKA.

## Materials and methods

### Study design

This was a single-center, retrospective study. All data acquisition and analyses were performed with the approval of our institutional review board (AJOUIRB-DB-2023-073). The data were collected and analyzed in March 2023 after the approval of our institutional review board. Among the patients who received primary UKA at our institution between March 2002 and March 2012, patients who had knee BS at least five years after UKA were included. Patients with inflammatory arthropathy, post-traumatic OA, a history of previous knee operations, and ligament deficiencies were excluded (Fig 1). A single senior surgeon performed all surgeries. BS results of the knee joints were analyzed with the progression of OA on radiograph images evaluated using the Kellgren–Lawrence grade (K-L grade) and Osteoarthritis Research Society International (OARSI) atlas score.

**Assessment of radiographic progression of OA in contralateral and patellofemoral compartments.** From our institution's picture archiving and communication system, plain radiographs taken at the first outpatient visit (usually between four and eight weeks) after

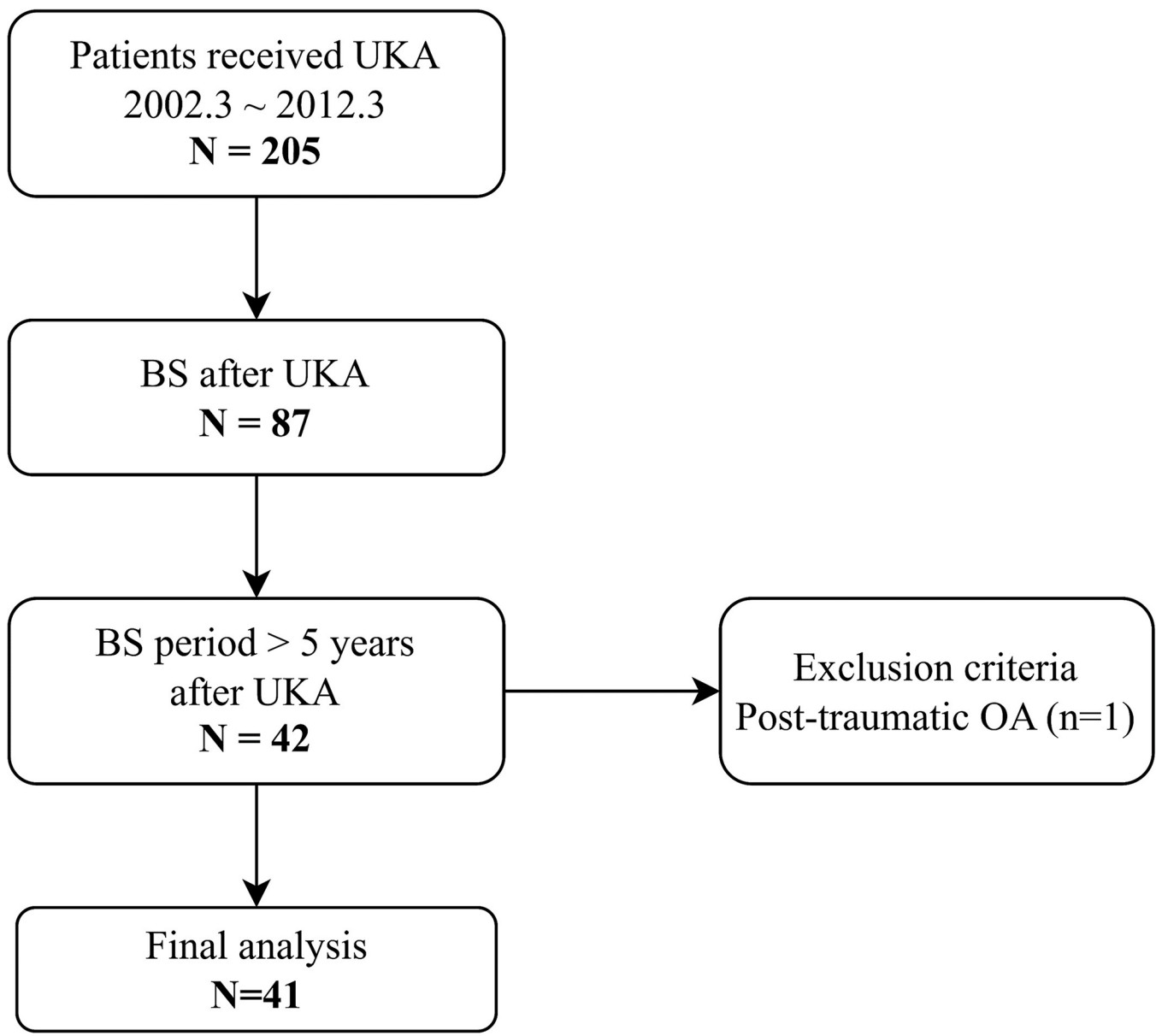

**Fig 1. Flow diagram shows the included patients that met the study criteria.** UKA, unicompartmental knee arthroplasty; BS, bone scintigraphy; OA, osteoarthritis.

surgery and those taken within two months of the BS were chosen for assessment. Radiographs taken for evaluating immediate postoperative status were not used for assessment due to the potential chance that foreign bodies such as dressing materials, staples, and drains could hamper the precise interpretation of OA severity in other compartments. Plain radiographs at follow-up routinely included weight-bearing knee anteroposterior, lateral, and Merchant radiographs used to assess OA severity in the contralateral tibiofemoral and patellofemoral compartments. The severity of OA was assessed using the K-L grade (scale ranging from zero to four) [24] and OARSI atlas score (ranging from zero to six) [25]. K-L grade one was defined as doubtful narrowing of joint space and possible osteophytic lipping; grade two, as definite osteophytes and possible narrowing of joint space; grade three, as moderate multiple

osteophytes, definite narrowing of joint space, and some sclerosis and possible deformity of bone ends; and grade four, as large osteophytes, marked narrowing of joint space, severe sclerosis, and definite deformity of bone ends [24]. The OARSI atlas score assessed joint space and osteophyte formation independently and scored each between zero to three (zero = none; one = mild; two = moderate; and three = severe), giving a total summary score out of six [26]. OA progression was defined as a K-L grade of one or higher and an OARSI atlas score of two or higher and was classified as "definitely progressed" (when the K-L grade was greater than one and the OARSI atlas score was greater than two), "possibly progressed" (when the K-L grade was equal to one and the OARSI atlas score was equal to two), or "not progressed" (when the K-L grade was lesser than one and the OARSI atlas score was lesser than two) [27, 28].

**BS for both knee joints.** Bone scan images of both knees (anteroposterior, posteroanterior, and lateral views) were obtained on a gamma camera (ORBITER, Siemens, Germany) 4 h after injection of $^{99m}$Technetium-labelled 2,3-di carboxy propane-1, 1-diphosphonate ($^{99m}$Tc-DPD) (20mCi). Positive BS uptake was defined as uptake on either the femoral or tibial side of the contralateral compartment or the patellar or trochlear side of the patellofemoral compartment. BS uptake was graded according to the intensity of bone scintigraphic radiolabel retention (zero to two: zero = normal; one = mild; and two = intense) [29].

**Statistical analysis.** Statistical analysis was performed using R, version 3.6.3. Spearman correlation analysis was used to assess the relationship between BS uptake and radiographic OA progression. Using Spearman correlation analysis, we also assessed the relationship between the BS uptake grade and radiographic OA progression grade. The level of significance was considered as a value of $p < 0.05$. Each BS image and radiograph were examined by two observers (SL, JML) in a blinded fashion, and intra-observer and inter-observer errors were evaluated using intra-class correlation coefficients (ICC). Reliability was considered poor when the ICC was < 0.40, fair when the ICC was 0.41–0.59, good when the ICC was 0.60–0.74, and excellent when the ICC was greater than 0.75 [30].

## Results

Forty-one patients with a knee joint BS evaluated at least five years after UKA were included (Table 1) in this study. In the contralateral compartment, OA progressed in 21 patients according to the K-L grade and in 15 patients according to the OARSI atlas score. In the patellofemoral compartment, 14 patients showed progression using the K-L grade, while 13 showed OA progression using the OARSI atlas score (Table 2). BS was performed at 98.5 ± 26.0 months after UKA, and uptake was observed in 16 patients in the contralateral and 26 in the patellofemoral compartments (Table 3).

**Table 1. Demographic details of the patients.**

| Demographic data | Patient group |
|---|---|
| Number of patients | 41 |
| Sex (Male) (%) | 13 (31.7%) |
| Mean age (years) | 64.6 ± 6.0 |
| Involved knee (right) (%) | 19 (46.3%) |
| Type of UKA (medial) (%) | 34 (82.9%) |
| Type of implant (fixed) (%) | 38 (92.7%) |
| Mean bone scan follow-up (months) | 98.5 ± 26.0 |

UKA, Unicompartmental knee arthroplasty

**Table 2. Changes in radiographic appearance of the contralateral tibiofemoral compartment and patellofemoral joint according to the K-L grade and OARSI score.**

| Condition | Contralateral | | Patellofemoral | |
|---|---|---|---|---|
| | K-L grade | OARSI score | K-L grade | OARSI score |
| Definitely worse | 4 | 3 | 2 | 4 |
| Probably worse | 17 | 12 | 12 | 9 |
| Same | 20 | 26 | 27 | 28 |

K-L, Kellgren-Lawrence; OARSI, Osteoarthritis Research Society International

**Table 3. Results of bone scan uptake grade in bone scintigraphy after unicompartmental knee arthroplasty.**

| BS uptake | CL | PF |
|---|---|---|
| Grade 0 | 25 | 15 |
| Grade 1 | 14 | 22 |
| Grade 2 | 2 | 4 |

BS, bone scintigraphy; CL, contralateral tibiofemoral joint; PF, patellofemoral joint

**Table 4. Correlation analysis between bone scan uptakes and OA progression.**

| Correlation analysis | Spearman's rho | p-value |
|---|---|---|
| BS (CL) vs. OA progression | | |
| K-L grade (CL) | -0.120 | 0.457 |
| OARSI (CL) | -0.089 | 0.582 |
| BS (PF) vs. OA progression | | |
| K-L grade (PF) | -0.094 | 0.762 |
| OARSI (PF) | 0.135 | 0.399 |

BS, bone scintigraphy; OA, osteoarthritis; CL, contralateral tibiofemoral joint; PF, patellofemoral joint

Correlation analysis results showed no correlation between BS uptake and radiographic OA progression assessed by the K-L grade and OARSI atlas score in the contralateral ($p = 0.457$, 0.582) and patellofemoral compartments ($p = 0.762$, 0.399) (Table 4). Moreover, after analyzing the correlation between radiographic OA progression and BS uptake grade, statistical significance was not found in the contralateral ($p = 0.426$, 0.639) and patellofemoral compartments ($p = 0.948$, 0.607) (Table 5).

K-L grade and OARSI atlas score were significantly correlated with each other in both contralateral (rho = 0.600, $p < 0.001$) and patellofemoral compartments (rho = 0.574, $p < 0.001$), indicating OA progression. When the contralateral compartment and patellofemoral joint were analyzed for correlation, the K-L grade and OARSI atlas score showed a significant correlation between OA progression in the contralateral (rho = 0.443, $p = 0.004$) and patellofemoral compartments (rho = 0.414, $p = 0.007$). However, BS uptake in the contralateral compartment was not significantly correlated with uptake in the patellofemoral compartment ($p = 0.077$).

The study results indicated good to excellent consistency in the measurements taken, as the ICC for measurement was in the range of 0.870–0.983 for intra-observer reliability and 0.708–0.829 for inter-observer reliability (Table 6).

**Table 5. Correlation analysis between bone scan uptake grades and OA progression grades.**

| Correlation analysis | Spearman's rho | p-value |
|---|---|---|
| BS (CL) vs. OA progression | | |
| K-L grade (CL) | -0.128 | 0.426 |
| OARSI (CL) | -0.076 | 0.639 |
| BS (PF) vs. OA progression | | |
| K-L grade (PF) | 0.011 | 0.948 |
| OARSI (PF) | -0.083 | 0.607 |
| K-L grade vs. OARSI | | |
| CL | 0.600 | **<0.001** |
| PF | 0.574 | **<0.001** |
| CL vs. PF | | |
| K-L grade | 0.443 | **0.004** |
| OARSI | 0.414 | **0.007** |
| BS | 0.279 | 0.077 |

BS, bone scintigraphy; OA, osteoarthritis; CL, contralateral tibiofemoral joint; PF, patellofemoral joint

**Table 6. Intra-observer and inter-observer ICCs of variable measurement.**

| | BS (CL) | BS (PF) | K-L (CL) | K-L (PF) | OARSI (CL) | OARSI (PF) |
|---|---|---|---|---|---|---|
| Intra-observer ICC | 0.870 | 0.970 | 0.924 | 0.880 | 0.983 | 0.973 |
| Inter-observer ICC | 0.826 | 0.829 | 0.804 | 0.728 | 0.803 | 0.708 |

BS, bone scintigraphy; CL, contralateral tibiofemoral joint; PF, patellofemoral joint; K-L, Kellgren-Lawrence; OARSI, Osteoarthritis Research Society International; ICC, intraclass correlation coefficient

## Discussion

In this study, we evaluated whether OA progresses to other compartments in patients who undergo UKA and if increased uptake on BS is associated with OA progression in other compartments after UKA. Our results showed radiographic progression of OA after UKA in 15–21 patients (36.6–51.2%) in the contralateral compartment and 13–14 patients (31.7–34.1%) in the patellofemoral compartment over a minimum follow-up period of five years. There was no significant correlation between radiographic OA progression and BS uptake, nor between radiographic OA progression grade and BS uptake grade.

Several results have been reported regarding OA progression after UKA; Berger et al. [31] concluded that radiographic OA progression was observed in approximately 50% of the patellofemoral and 57% of the contralateral compartments during more than 10 years. Walton et al. [32] reported that definite OA progression was seen in 18–34% of the retained compartments after reviewing 32 lateral UKA cases assessed at five years. According to Misir et al. [27], after an average of 7.41 years of observation following medial UKA, OA progression was observed in 34.9% of the lateral and 45.2% of the patellofemoral compartments andfound no correlation between patellofemoral OA progression and outcome, but lateral OA progression did affect clinical outcomes. Similarly, several studies have reported that mild-to-moderate patellofemoral OA does not affect UKA outcomes, and the indications for UKA have been expanded to include cases with OA and malalignment of the patellofemoral joint [33–36]. The radiographic evaluation in our study showed OA progression of 36.6–51.2% in the contralateral and 31.7–

34.1% in the patellofemoral compartments during a mean follow-up of 98.5 ± 26.0 months, similar to previous results. Additionally, Misir et al. [27] reported no correlation between patellofemoral and contralateral OA progression after UKA. However, our study found a significant correlation between the two as measured by radiography in both grading systems, but no correlation with BS uptake was observed. It is known that several factors are associated with the progression of OA after UKA [2, 27, 37, 38]. But there have been few reports regarding relationship OA progression between contralateral and patellofemoral compartment [27]. This study only included patients who had BS after UKA surgery, and the small number of patients may have contributed to the differing results compared to the previous study by Misir et al. Nevertheless, in the future, it is believed that this study can serve as a reference for understanding the relationship of OA progression in the remaining two compartments after UKA.

Most BS in previous studies has been obtained with $^{99m}$Tc-MDP [16, 17, 21, 39–43], but our institution uses $^{99m}$Tc-DPD for BS for both knee joints. There was no established superiority between the two agents when comparing bone imaging using each of the agents. Previous studies have shown comparable performance in favor of $^{99m}$Tc-MDP in various pathologic conditions, including metastasis, rheumatoid arthritis, metabolic diseases, and bone fractures [44, 45]. On the other hand, Buell et al. argued superior bone-to-soft tissue ratios of $^{99m}$Tc-DPD over $^{99m}$Tc-MDP for both high- and low-uptake bone [46]. Concerning bone imaging in osteoarthritic conditions, Lee et al. designed an animal (beagle dog) OA model to compare the effect of both agents on BS image quality and time; their results demonstrated that both agents showed similar effects on radioactive uptake ratio and image quality; furthermore, $^{99m}$Tc-DPD was more efficient than $^{99m}$Tc-MDP in reducing the overall time of scintigraphy [47].

Furthermore, several studies have shown that BS can provide valuable insights into the disease process of OA. McCrae et al. [17] first reported that BS could detect different scintigraphic abnormalities reflecting various aspects of OA. Some studies have found that BS can help identify early changes in OA and correlate them with its severity [16, 22, 23]. Park et al. [23] showed that BS uptake correlated with articular cartilage degeneration in a histologic study. From a biochemical perspective, OA-related biomarkers such as serum and synovial fluid, cartilage oligomeric matrix protein, and bone sialoprotein have shown a certain correlation with bone scintigraphic findings [13, 48]. In a comparison study using MRI, increased uptake on BS showed good agreement with MRI-detected subchondral lesions [39, 42]. Additionally, there is evidence that BS can predict OA progression [15, 40, 43]; however, some studies have shown that the K-L grading system is better at predicting OA progression compared to BS [40, 43].

But few studies have investigated BS results after UKA. According to Mandegaran et al. [49], BS had low sensitivity and specificity in evaluating aseptic loosening and infections compared to single-photon emission computed tomography. Wong et al. [41] reported that BS was not significantly helpful in identifying loosening or infection in painful mobile-bearing UKA, but it could identify changes in the contralateral and patellofemoral compartments after UKA. In knee OA, the compartment-specific localization and intensity of BS retention are also associated with the localization and severity of radiographic OA [16]. In patients who have undergone UKA surgery, X-ray grading may be limited by post-operative implant interference, but BS uptake can be measured without interference from the implant, thereby having a superior diagnostic ability in post-operative conditions. However, the results of this study showed that there was no correlation between the presence or grading of BS uptake and OA progression, contrary to our hypothesis.

This study had several limitations that should be acknowledged. First, it was retrospectively designed with a relatively small patient group (41 patients), and only mid-term follow-up results could be obtained. Second, OA progression after UKA is affected by the position and

alignment of the implant, but this was not evaluated in the study. Third, patient-reported outcomes were not assessed in our study, as the operated compartment primarily has the greatest impact on patient-reported outcomes. Fourth, we included medial and lateral compartment UKA in the study regardless of bearing type, which may cause selection bias. However, to the best of our knowledge, this study was the first to evaluate the value of BS in OA progression in the retained compartments after UKA. Our analysis showed a correlation in OA progression between contralateral tibiofemoral and patellofemoral compartments after UKA, which has not been fully explored in the literature. A large-scale, well-designed prospective study may provide more solid evidence on these topics.

## Conclusion

Following UKA, OA progresses in the retained contralateral tibiofemoral and patellofemoral compartments over a minimum five-year follow-up period. BS is ineffective in assessing the progression of OA in these compartments.

## Supporting information

**S1 Dataset.**
(XLSX)

## Author Contributions

**Conceptualization:** Sumin Lim, Do Young Park, Jun Young Chung.

**Data curation:** Sumin Lim, Tae Hun Kim, Do Young Park, Jong Min Lee, Jun Young Chung.

**Formal analysis:** Sumin Lim, Tae Hun Kim, Do Young Park, Jong Min Lee, Jun Young Chung.

**Investigation:** Sumin Lim, Tae Hun Kim, Do Young Park, Jong Min Lee, Jun Young Chung.

**Methodology:** Sumin Lim, Tae Hun Kim, Do Young Park, Jun Young Chung.

**Project administration:** Jun Young Chung.

**Resources:** Do Young Park, Jong Min Lee, Jun Young Chung.

**Supervision:** Do Young Park, Jun Young Chung.

**Validation:** Sumin Lim, Tae Hun Kim, Do Young Park, Jong Min Lee, Jun Young Chung.

**Visualization:** Jun Young Chung.

**Writing – original draft:** Sumin Lim.

**Writing – review & editing:** Sumin Lim, Jun Young Chung.

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
