## [Decision Letter · Decision Letter 0]

14 Jun 2023

PONE-D-23-11014Can Bone Scintigraphy Reflect the Progression of Osteoarthritis after Unicompartmental Knee Arthroplasty?PLOS ONE

Dear Dr. CHUNG,

Thank you for submitting your manuscript to PLOS ONE. After careful consideration, we feel that it has merit but does not fully meet PLOS ONE’s publication criteria as it currently stands. Therefore, we invite you to submit a revised version of the manuscript that addresses the points raised during the review process.

We look forward to receiving your revised manuscript.

Kind regards,

Amit Joshi, MD

Academic Editor

PLOS ONE

Journal Requirements:

**Additional Editor Comments:**

Dear Author.

Kindly revise your manuscript as per commented by our reviewer.

Reviewers' comments:

Reviewer's Responses to Questions

**Comments to the Author**

1. Is the manuscript technically sound, and do the data support the conclusions?

Reviewer #1: Yes

Reviewer #2: Yes

2. Has the statistical analysis been performed appropriately and rigorously? 

Reviewer #1: Yes

Reviewer #2: Yes

3. Have the authors made all data underlying the findings in their manuscript fully available?

Reviewer #1: Yes

Reviewer #2: Yes

4. Is the manuscript presented in an intelligible fashion and written in standard English?

Reviewer #1: Yes

Reviewer #2: Yes

5. Review Comments to the Author

Reviewer #1: Abstract: Professionally written abstract

Introduction:

1. The statement made in line 50-53 requires citation.

2. The statement made in line 64 requires citation.

3. Clear study background and good flow of writing

4. Clear objectives and study hypothesis

Methodology:

1. Clear study selection criteria

2. The statement made in line 96-101 requires citation, i.e., [18]

3. Line 104-109 can be incorporated into a single sentence with citation or provide citation at the end of following sentence too, i.e. [21-22]

4. Both sentences, from line 111-117, requires citation

Result:

1. Flow of participants using a flow-chart would be better

2. Information regarding the occurrence of aseptic implant loosening would be interesting

Discussion:

1. Avoid repetition (similarly written in introduction)

2. line 162-68 can be deleted to establish a flow

3. line 183-85: present study observed significant correlation between retained compartment OA progression compared to that reported by Misir et al. As this is the significant finding, further reasoning of this particular outcome is required. It can either be achieved by critical evaluation of Misir et al. study or identifying the reasons behind such significant correlation in the present study.

4. Discussion section needs improvement. Author should give more effort in establishing the strengths and implications of study findings. It is also important to compare or contrast with the findings of previous study based on their methodological quality.

Reviewer #2: It is good to know with this study that OA in other compartment and patellofemoral OA progression occurs after few years of UKA. On top of that from this study it is known that patellofemoral OA doesnot alter clinical outcome. So UKA indications expanded to patient with Patellofemoral OA too. But it is better to search for better non invasive procedures prospectively to see OA progression with larger samples. Anyway I congratulate authors who have given their great efforts in preparing this article with good methods and methodology, adequate sample sizes, valid statististical tools and scientific analysis. I recommend prospective study with larger sample sizes performed by single surgeon in definite age group which might remove biasness regarding surgeon related mechanical failure cases.

6. PLOS authors have the option to publish the peer review history of their article (what does this mean?). If published, this will include your full peer review and any attached files.

Reviewer #1: **Yes: **Subhash Regmi

Reviewer #2: **Yes: **Shrawan Kumar Thapa

While revising your submission, please upload your figure files to the Preflight Analysis and Conversion Engine (PACE) digital diagnostic tool, https://pacev2.apexcovantage.com/. PACE helps ensure that figures meet PLOS requirements. To use PACE, you must first register as a user. Registration is free. Then, login and navigate to the UPLOAD tab, where you will find detailed instructions on how to use the tool. If you encounter any issues or have any questions when using PACE, please email PLOS at figures@plos.org. Please note that Supporting Information files do not need this step.<quillbot-extension-portal></quillbot-extension-portal>

---

## [Author Response · Author response to Decision Letter 0]

23 Jun 2023

I have uploaded the 'Response to reviewers' File separately, thank you.

---

## [Editor Report · Decision Letter 1]

3 Jul 2023

Can Bone Scintigraphy Reflect the Progression of Osteoarthritis after Unicompartmental Knee Arthroplasty?

PONE-D-23-11014R1

Dear Dr. CHUNG,

We’re pleased to inform you that your manuscript has been judged scientifically suitable for publication and will be formally accepted for publication once it meets all outstanding technical requirements.

Kind regards,

Amit Joshi, MD

Academic Editor

PLOS ONE

Additional Editor Comments (optional):

thank you for adressing the comments made by our reviewer. the manuscript is provisionally accepted.

Reviewers' comments:

<quillbot-extension-portal></quillbot-extension-portal>

---

## [Editor Report · Acceptance letter]

5 Jul 2023

PONE-D-23-11014R1 

Can Bone Scintigraphy Reflect the Progression of Osteoarthritis after Unicompartmental Knee Arthroplasty? 

Dear Dr. Chung:

I'm pleased to inform you that your manuscript has been deemed suitable for publication in PLOS ONE. Congratulations! Your manuscript is now with our production department. 

Kind regards, 

on behalf of

Professor Amit Joshi 

Academic Editor

PLOS ONE